

1          **33**

# Mapping and quantifying isomer sets of hydrocarbons (≥C₁₂) in diesel fuel, lubricating oil and diesel exhaust samples using GC×GC-ToFMS

**Mohammed S. Alam[1], Zhirong Liang[2], Soheil Zeraati-Rezaei[3]**

**Christopher Stark[1], Hongming Xu[3], A. Rob MacKenzie[1¥] and**

**Roy M. Harrison[1*†]**

**[1] Division of Environmental Health and Risk Management, School of Geography, Earth and Environmental Sciences, University of Birmingham, Edgbaston, Birmingham B15 2TT, United Kingdom**

**[2] School of Energy and Power Engineering, Beihang University Beijing, 100191 China**

**[3] School of Mechanical Engineering, University of Birmingham, Edgbaston Birmingham, B15 2TT, United Kingdom**

[*] To whom correspondence should be addressed.

Tele: +44 121 414 3494; Fax: +44 121 414 3709; Email: r.m.harrison@bham.ac.uk

[†]Also at: Department of Environmental Sciences / Center of Excellence in Environmental Studies, King Abdulaziz University, PO Box 80203, Jeddah, 21589, Saudi Arabia

[¥] Also at: Birmingham Institute of Forest Research, University of Birmingham, B15 2TT





**ABSTRACT**
Many environmental samples, including water, soils, sediments and airborne particles and vapours
contain complex mixtures of hydrocarbons, often deriving from crude oil either before or after
fractionation into fuels, lubricants and feedstocks.    Comprehensive 2D Gas Chromatography- Time-
of-Flight Mass Spectrometry (GC×GC-ToFMS), offers a very powerful technique separating and
identifying many compounds in complicated hydrocarbon mixtures. However, quantification and
identification of individual constituents at high ionization energies would require hundreds of
expensive (if available) standards for calibration. Although the chemical structure of hydrocarbons
does matter for their environmental impact and fate, strong similarities can be expected for
compounds having very similar chemical structure and carbon number. There is, therefore, a clear
benefit in an analytical technique which is specific enough to separate different classes of compounds,
and to distinguish homologous series, whilst avoiding the need to handle each isomer individually.
Varying EI (electron impact) ionization mass spectrometry significantly enhances the identification
of individual isomers and homologous compound groups, which we refer to as 'isomer sets'.
Advances are reported in mapping and quantifying isomer sets of hydrocarbons ($\geq C_{12}$) in diesel fuel,
lubricating oil and diesel exhaust emissions. Using this analysis we report mass closures of *ca.* 90%
and 75% for diesel fuel and lubricating oil.





# 1. INTRODUCTION

Crude oil contains a highly complex mixture of chemical constituents, mainly hydrocarbons ($C_4$-$C_{55}$) (Riazi, 2005). There are many reports of crude oil entering the environment through spillage, or deliberate release (Gertler et al., 2010). Most crude oil is treated and fractionated in order to produce fuels and lubricants for use in transport and combustion applications, and as feedstocks for the chemical industry (Riazi, 2005). All of these uses have a potential to contaminate the environment. Understanding fates, pathways, and effects of contamination requires chemical analysis and detailed interpretation of resulting data. Much of the chemical complexity of oil derives from the large numbers of straight and branched chain and cyclic hydrocarbon isomers for a given carbon number (Goldstein et al., 2007). Hence, analytical methods are required that can discriminate structurally similar sets of isomers in complex media.

Application of conventional gas chromatographic methods to oils and oil-derived samples was for many years severely limited by the poor separation capability of one-dimensional chromatography, due to the near-continuous range of physicochemical properties of hydrocarbons. Thus, typically 90% of the hydrocarbon content of the sample is present in the unresolved complex mixture (UCM), creating a large hump in the chromatogram (Fraser et al., 1998; Schauer et al., 1999). The advent of two-dimensional gas chromatography, which provides enhanced separating capability due to the orthogonal separation by two capillary columns of different stationary phases, has transformed the problem by resolving the UCM into many thousands of individual compound peaks. The two columns are connected in series by a modulator which is employed to provide focusing of the primary column eluent (Liu et al., 1991; Phillips et al., 1995). Large amounts of data are produced due to the large





number of compounds separated. This technique generates very large volumes of mass spectral data,
although it is often generic information which is required in order to compare the main compositional
attributes of samples, rather than detailed identification of many compounds.   Use of a flame
ionisation detector has the advantage of allowing generic quantification of any part of the
chromatogram in terms of the carbon mass contained within it, but identification of specific chemical
constituents with this detector can only be achieved on the basis of retention times which, in a very
complex two-dimensional chromatogram and set-up-dependent chromatogram, are laborious to
assign objectively. Mass spectrometric detection, especially when employing both low and high
ionisation energies adds a third analytical dimension with the ability to overcome the problem of
compound identification (Alam et al., 2016a), but has not generally been applied to generic
quantification of compound groups within complex samples.   In this study, we show that time-of-
flight mass spectrometric detection can be used not only to identify and quantify individual chemical
constituents within the chromatogram, but can also be used to quantify generic groups of compounds.

Motor vehicles are a major source of organic carbon in the environment, and the majority of the fine
particulate matter (PM) emitted is carbonaceous, directly emitted as primary organic aerosol (POA)
or formed as secondary organic aerosol (SOA) (Jimenez et al., 2009). A substantial fraction of the
POA in vehicle emissions has been shown to be semi-volatile under atmospheric conditions
(Robinson et al., 2007; May et al., 2013), and is mainly comprised of aliphatic species in the carbon
number range between $C_{12}$–$C_{35}$, with effective saturation concentrations (C*) between 0.1 and $10^3$ µg
$m^{-3}$ (Robinson et al., 2007; Weitkemp et al., 2007).   The semi volatile organic compound (SVOC)
composition of lubricating oil has been reported to be dominated by branched, cyclic and straight
alkanes (≥80%), with the largest contribution from cycloalkanes (≥27%) (Worton et al., 2014; Sakurai



et al., 2003).

Previous research has used a limited range of tracer compounds, or homologous series, for the
quantification of emissions, considering representative species that can be distinguished from the bulk
of the mass, typically involving analysis of the n-alkanes, polycyclic aromatic hydrocarbons (PAH),
hopanes and steranes (Schauer et al., 1999; 2002) each of which represent only a small fraction of the
total mass or number of compounds emitted and might lead to underestimation of the importance of
lubricating oil as a source of SOA (Brandenberger et al., 2005; Fujita et al., 2007).    Although some
studies have utilized soft ionization to analyse diesel fuel at a molecular level, (Briker et al., 2001;
Eschner et al., 2010; Amirav et al., 2008) very few studies have analysed lubricating oil at a molecular
level that includes the analysis of SVOCs (Worton et al., 2015; Reddy et al., 2012).    In order to
address the problems of coelution of constituents of the UCM, Worton et al. (2014) and Isaacman et
al. (2012) utilized gas chromatography coupled with vacuum ultraviolet ionization mass spectrometry
(GC/VUV-MS) to study the constitutional isomers present in lubricating oil and diesel fuel,
respectively, and in a standard crude oil from the Gulf of Mexico (Worton et al., 2015).

Using GC×GC allows compounds of similar chemical structure to be classified into distinct groups
in ordered chromatograms based on their volatility and their polarity, providing information that aids
identification.    Dunmore et al. (2015) recently grouped low molecular weight ($\leq C_{12}$) hydrocarbons
in atmospheric samples by carbon number and functionality using GC×GC. They reported the
grouping of $C_6$–$C_{13}$ aliphatics and $C_2$–$C_4$ substituted monoaromatics, combining the area of all the
peaks contained within their selected areas.



In our study, two dimensional Gas Chromatography Time-of-Flight-Mass Spectrometry (GC×GC-
ToF-MS) (Adahchour et al., 2008; Alam et al., 2013; Alam et al., 2016b) was utilised and combined
with an innovative quantification methodology based on total ion current (TIC) signal response to
provide identification and quantification for the compound classes within typical diesel fuel, engine
lubricant and engine emissions (gas and particulate phases), providing    a near complete mass closure
for diesel fuel and engine lubricant and analyses of diesel engine exhaust composition.

**2.       EXPERIMENTAL**
**2.1      Sampling**
Diesel fuel, engine lubricating oil and gas/particulate diesel exhaust emission samples were analysed
using GC×GC-ToF-MS. Briefly, 1 µL of diesel fuel (EN 590-ultra low sulfur diesel, Shell, UK) was
diluted (1:1000) in dichloromethane (DCM) and injected onto a stainless steel thermal adsorption
tube, packed with 1 cm quartz wool, 300 mg Carbograph 2TD 40/60 (Markes International), for
analysis on the thermal desorber (TD) coupled to the GC×GC-ToF-MS. The EN 590 ultra-low sulphur
diesel fuel is representative of the standardized ultra-low sulphur content fuel (<10 mg kg$^{-1}$ or ppm)
that is widely utilised in the UK and Europe (Ref: EU directive 2009/30/EC). 1 µL of engine lubricant
(fully synthetic, 5W30, Castrol, UK) was diluted (1:1000) in DCM and directly injected into the gas
chromatographic column, as the high molecular weight constituents found in the lubricating oils
would not efficiently desorb into the GC column from the adsorption tubes.
Details of the engine exhaust sampling system are given elsewhere (Alam et al. 2016c). Briefly,
adsorption tubes were used to collect gas phase constituents directly from diluted (1:50) diesel engine
exhaust, downstream of a polypropylene backed PTFE 47 mm filter (Whatman, Maidstone, UK) used
to collect and remove constituents in the particulate phase. The diesel engine emissions were diluted



(1:50) with cleaned compressed air using an in-house exhaust dilution system described elsewhere
(Alam et al., 2016c).    The temperature at the sampling point was $25 \pm 5$ °C. Samples were collected
for 30 min at a flow rate of 1.8 L min$^{-1}$. Details of the engine set up have been given elsewhere (Alam
et al., 2016c).    Samples were collected at steady state engine operating conditions at a load of 3.0
bar mean effective pressure (BMEP) and a speed of 1800 revolutions per minute (RPM) without a
diesel oxidation catalyst (DOC) and diesel particulate filter (DPF).

**2.2    Instrumentation**
Adsorption tubes were desorbed using TD (Unity 2, Markes International, Llantrisant, UK) and
samples were subsequently analysed using a gas chromatograph (GC, 7890B, Agilent Technologies,
Wilmington, DE, USA) equipped with a Zoex ZX2 cryogenic modulator (Houston, TX, USA). The
primary column (first separation dimension) was equipped with a SGE DBX5, non-polar capillary
column (30 m, 0.25 mm ID, 0.25 μm – 5% phenyl polysilphenylene-siloxane). The secondary, more
polar column (second separation dimension) was equipped with a SGE DBX50 (4.0 m, 0.1 mm ID,
0.1 μm – 50% phenyl polysilphenylene-siloxane), situated in a secondary internal oven. The GC×GC
was interfaced with a BenchTOF-Select, time-of-flight mass spectrometer (ToF-MS, Markes
International, Llantrisant, UK), with a scan speed of 50 Hz, covering the mass/charge range from 30
to 525 m/z. Electron impact ionisation energies on this ToFMS can be tuned between 10 eV and 70
eV, the former retaining the molecular ion, while the latter causes extensive fragmentation, but allows
comparison with standard library spectra.[6] Data were processed by using GC Image v2.5 (Zoex
Corporation, Houston, US).






**2.3      Standards & Chromatography Methodology**
Nine   deuterated   internal   standards   namely,   dodecane-$d_{26}$,   pentadecane-$d_{32}$,   eicosane-$d_{42}$,
pentacosane-$d_{52}$,  triacontane-$d_{62}$,  biphenyl-$d_{10}$,  $n$-butylbenzene-$d_{14}$,  $n$-nonylbenzene-2,3,4,5,6-$d_5$
(Chiron AS, Norway) and $p$-terphenyl-$d_{14}$ (Sigma Aldrich, UK) were utilised for quantification.
Natural standards included 24 $n$-alkanes ($C_{11} - C_{34}$), phytane and pristane (Sigma Aldrich, UK), 16
$n$-alkylcyclohexanes ($C_{11} - C_{25}$ and $C_{26}$), 5 $n$-alkylbenzenes ($C_4$, $C_6$, $C_8$, $C_{10}$ and $C_{12}$), cis- and trans-
decalin, tetralin, 4 alkyltetralins (methyl-, di-, tri- and tetra-), 4 $n$-alkyl naphthalenes ($C_1$, $C_2$, $C_4$ and
$C_6$) (Chrion AS, Norway) and 16 USEPA polycyclic aromatic hydrocarbons (Thames Restek UK Ltd).
These standards were chosen in order to cover as much of the overall chromatogram as possible. The
chromatography  methodology  of  the  analysis  of  the  adsorption  tubes,  lubricating  oil  and
gas/particulate phase samples is discussed in Section S1 in the Supplementary Information.


**2.4      Grouping of Chromatographically Resolved Compounds**
Compounds belonging to the same chemical group in a mixture possess similar physicochemical
properties. This facilitates identification when separated according to these physical and chemical
properties. Diesel fuels, diesel emissions and lubricating oils have been shown to consist of a limited
number of compound classes, but an enormous number of individual components within a class.

In this study we use GC×GC coupled to variable ionisation ToF-MS to map and quantify isomer sets
previously unresolved in the UCM. Conventional electron ionisation at 70 eV imparts a large amount
of excess energy causing extensive fragmentation, with a tendency to generate similar mass spectra.
Thus for example the isomeric alkanes all exhibit the same m/z 43, 57, 71, 85, 99 patterns, thus



obscuring the match with the NIST library and making identification from the mass spectrum very
difficult. To address this issue, a lower ionization energy (10-14eV) was also employed in our study
so that the organic compounds are ionized with minimal excess internal energy and thus less
fragmentation, hence retaining the distinct identity of the molecule with a much larger fraction of the
molecular ion (Alam et al., 2016a).    Running samples on the GC×GC with both low and high
ionization energy mass spectrometry results in a wealth of data for identification of compounds,
where 14 and 70eV mass spectra can be compared for a given species owing to the identical retention
times of the repeat runs. At low ionization energy the molecular ion is enhanced, while still retaining
some fragmentation, while at high ionization energy the mass fragmentation patterns of a species can
be compared directly to mass spectral libraries. This allows easier identification of unknown
compounds.

Our recent work exploited soft ionisation (14 eV) to identify a large number of isomers, demonstrating
the ability to separate and identify individual alkanes (normal, branched and cyclic) with specific
carbon numbers, based on their volatility and polarity (Alam et al., 2016a).    In this study we expand
our previous qualitative analysis and separate the alkane series (as well as other homologous series)
into isomer sets containing the same carbon number. Individual alkanes that were identified as having
different molecular ions (i.e. different carbon number) to the *n*-alkane within the area of the
chromatogram were included in their appropriate adjacent (usually *n*±1) area; for example, some
dimethyl isomers can be shifted by ~100 delta-Kovats (~1 carbon number), whereas trimethyl- and
tetramethyl isomers have been reported to be shifted by ~150 and ~200 delta-Kovats. This has been
completed for all the homologous series reported in this study. The grouping of the alkanes according
to their respective carbon numbers is shown in Figure 1, where the least polar compounds (fast eluting



peaks in the second dimension) are the alkanes, increasing in carbon number as the retention time in
the first dimension increases.

This methodology was expanded to more polar homologous series including monocyclic alkanes,
bicyclic alkanes, tricyclic alkanes, tetralins/indanes, monocyclic aromatics, bicyclic aromatics and
alkyl-biphenyls. Like the alkanes, a significant problem in creating the boundaries of groups is the
overlapping of one carbon number group into another. Identifying each individual compound in this
case (as with the alkanes above) would be resource and time intensive and so carefully constructed
'Computer Language for Identification of Chemicals' (CLIC) qualifiers were created and utilised in
order to match peaks and their mass fragmentation patterns. A CLIC qualifier is an expression in a
computer language that allows users of chromatographic software to build rules for selecting and
filtering peaks using retention times and mass fragmentation patterns (Reichenbach et al., 2005).
This was exploited to identify specific compounds belonging to a compound class and a polygon
selection tool within the GC Image software was drawn around this section of the chromatogram
(coloured polygons shown in Figure 1). Any overlap in the graphics was accounted for by forcing
peaks to belong to one compound class over another via strict mass fragment and molecular ion
selection tools. Examples of CLIC expressions utilised for identifying compound classes are included
in the Supplementary Information (Section S2). An example of a selected ion chromatogram with a
specific CLIC expression is shown in Figure 2, for $C_6$-substituted monocyclic aromatics, with their
corresponding 70eV and 12eV mass spectra. The characteristic 70eV mass fragments at m/z 92, 105,
119, 133 signify cleavage of the C-C bond next to the benzene ring. The 12eV mass spectra, however,
produce poor characteristic fragment ions, but a prominent molecular ion (162) and m/z 92 signifying
the overall mass of the molecule and the benzene ring (Ph-$CH_2^+$), respectively. In effect, the polygons





mark out sets of isomeric compounds having the same empirical formula and shared structural
elements; the sets appear to intersect each other in the two-dimensional chromatogram space, but
compounds in the intersecting regions are assigned uniquely to a class using a third, mass
spectrometric, data dimension (i.e. stringent mass fragmentation patterns). The resulting isomer sets
are more chemically and environmentally meaningful than the raw polarity/volatility assignment
from the chromatography. This approach was completed independently for diesel, lubricating oil and
gas/particulate phase emissions to ensure the polygon boundaries applicability and reproducibility of
retention times and mass fragments. Results indicated that isomers within the constructed polygon
boundaries possessed identical retention times and interpretable mass spectra for all differing samples.

The total ion current within each polygon was integrated and the isomer set abundance was estimated
using the response ratio of the closest structurally-related deuterated standard to the corresponding
compound class natural standard with the same carbon number (usually within the polygon). This
methodology has an uncertainty of approximately 24% and is discussed in detail in the Supplementary
Information (Section S3). The potential of scaling ion current to molar quantity for $<C_{25}$ and mass for
compounds $>C_{25}$ is also discussed in the Section S4.

**3.   RESULTS AND DISCUSSION**
**3.1       Analysis of Diesel Fuel**
The chromatography of the diesel fuel analysed by TD-GC×GC-ToF-MS is shown in Figure 1.
Compounds identified within the diesel fuel included: *n*-alkanes, branched alkanes (mono-, di-, tri-,
tetra- and penta-methyl), *n*-alkyl cycloalkanes, branched monocyclic alkanes, $C_1$-$C_{12}$ substituted
bicyclic alkanes, $C_1$-$C_4$ substituted tetralins and indanes, $C_3$-$C_{12}$ substituted monocyclic aromatics,



$C_1$-$C_3$ substituted biphenyls/acenaphthenes, $C_1$-$C_4$ substituted bicyclic aromatics, $C_1$-$C_2$ substituted
fluorenes (FLU), $C_1$-$C_2$ substituted phenanthrene/anthracenes (PHE/ANT) and unsubstituted PAH.
Representative mass spectra at 12eV and 70eV ionization are presented in the Supplementary
Information (Section S5). These compounds accounted for 93% of the total response (excluding the
siloxanes which derive from contaminants *i.e.* column bleed) which was equivalent to 90 % of the
mass injected. Therefore, out of the 8026 ng that was injected into the GC×GC, a mass of
approximately 7200 (± 1728) ng was accounted for. We suspect that a significant amount of the mass
that was unaccounted may be $<C_{10}$ and/or any unresolved peaks that we were unable to measure
and/or identify using our technique (see Supplementary Information, Section 6). The percentage
contribution of each compound class identified to the total mass accounted for is shown in Table 1.


Our results indicate that the majority of the diesel fuel consists of aliphatic compounds, with a low
aromatic content (~10%). Very few published studies exist elucidating the contribution of different
constituents in diesel fuel (Isaacman et al., 2012; Welthagen et al., 2007; Gentner et al., 2012). Most
studies focus on the characterisation of specific compounds within diesel fuels such as nitrogen
containing species (Wang et al. 2004), cyclic compounds (Edam et al., 2005), or to identify strengths
and weaknesses in analytical techniques (Frysinger et al., 1999). Recently, VUV ionization at 10 –
10.5 eV has been exploited to elucidate some of the structures within diesel fuel, by separating
components using GC (Isaacman et al., 2012). The authors report their observed mass of diesel fuel
as 73% aliphatic and 27% aromatic, broadly consistent with the results of this study. Up to 11% of
the observed mass fraction of diesel fuel was attributed to bicyclic alkanes, a factor of 2 larger than
observed in this study. Their observed mass fractions of cycloalkanes and benzene, however, are in





excellent agreement. The contribution of branched alkanes (*i*-alkanes) and linear *n*-alkanes to the total
mass of the alkanes was 39.1 and 23.1%, respectively. A significant proportion of the total mass
observed was attributed to alkanes (62%), a factor of 1.5 larger than reported by Isaacman et al. (2012).
However, the differences observed between diesel fuel analysed in this study and that reported by
Isaacman et al. (2012) is attributable to different fuel formulations and/or fuel source, as opposed to
analytical methods. Although not shown here, a significant number of alkane isomers were identified
for each carbon number using soft ionisation mass spectrometry, accounting for a total of ~200
alkanes across the $C_{11} - C_{30}$ range. The ratio of *i*-alkanes to *n*-alkanes sharply decreases after $C_{25,}$
indicating a reduced amount of mass represented by branched isomers present in diesel fuel for $>C_{25}$
alkanes, which could be related to the formulation process, or reflect the composition of the feedstock.

**3.2 Analysis of diesel engine emissions (gas phase)**
A GC×GC contour plot of the gas phase diesel exhaust emissions is shown in the Supplementary
Information (Figure S7). The observed chromatogram for the gas phase emissions looked extremely
similar to that of the diesel fuel chromatogram (Figure 1), suggesting that the majority of compounds
found in the gas phase emissions are of diesel fuel origin. All of the compounds found in the diesel
fuel were observed in the gas phase emissions, albeit with a reduced number of *i*-alkanes $>C_{20,}$ which
may signify efficient combustion of these high molecular weight compounds, or partitioning into the
particulate phase. The measured constituents of the gas phase diesel exhaust emissions are shown in
Table 1. Approximately 15 % of the total ion current (response, excluding siloxanes) was
unaccounted for. Table 1 illustrates the percentage mass of each compound class identified, in the 85%
of the response that was accounted for. As the total mass of the gas phase sample is unknown, a mass
for the remaining 15% of the total response cannot be estimated, as the individual components that



are unidentified will have different responses per unit mass. For example, 23.5% of the mass
identified was attributed to $C_4 – C_{12}$ alkyl substituted monocyclic aromatics and accounted for 9.7%
of the total ion current response; and 10.0% of the mass identified was bicyclic alkanes, representing
9.2% of the response.

Although the diesel fuel constituents present in the gas phase exhaust emissions broadly were
compositionally consistent with the fuel, there were significant differences observed in their relative
amounts. Of the total mass identified in the gas phase emissions, *n*-alkanes and *i*-alkanes represented
9.8% and 30.1%, respectively. These are factors of 2.4 and 1.3 lower than that for diesel fuel,
respectively; which may be due to preferred combustion of these compounds (Burcat et al., 2012).
Enhancements of monocyclic aromatics, monocyclic alkanes, bicyclic alkanes and bicyclic aromatics
were observed in the emissions, possibly due to them being intermediate species formed during the
combustion of larger molecules (Gentner et al., 2013), and unlikely to be a contribution from
lubricating oil as very little mass was attributed to compounds with $<C_{18}$ (see Section 3.3), in contrast
to previous studies (Worton et al., 2014).    A very limited number of oxygenates were also identified
(*e.g.* ketones (m/z 58, 72), carboxylic acids (m/z 60)), most probably combustion products of diesel
fuel, but representing a very small fraction of the total measured gas phase emissions (<1%). Gentner
et al. (2013) suggest that compounds such as alkenes, aromatics and oxygenates comprise ~30% of
the total measured gas phase emissions, in agreement with this study; however, they suggest that
these products are unlikely to contribute to primary organic aerosol (POA). We observe these
aromatic compounds in the particulate phase also, indicating a contribution.

**3.3      Analysis of Lubricating Oil**



A comprehensive analysis of base oil is presented elsewhere (Alam et al, 2016a) and a comparison
of different aged and fresh oils is reported in Liang et al. (2017 in prep). A brief account of analysis
of 5W30 synthetic lubricating oil is presented in this study. The analysis was conducted in exactly
the same way as the diesel fuel and gas phase diesel emissions as outlined in Section 2.4. In brief,
analysis was conducted using 12eV electron impact ionization energy mass spectrometry in order to
retain the molecular ion for the compounds analysed. This enabled the clear identification of specific
compounds with different carbon numbers from within a compound class. Molecular ions present in
the mass spectra enabled the grouping of isomers by carbon number, while the presence of the
characteristic mass fragments, presented in Table 1, were used to confirm the identity of the type of
hydrocarbon, see Supplementary Information, Section S8, for the representative mass spectra for
compounds presented in Table 1. Polygons were drawn around groups of compounds that possessed
the same molecular ion for a given compound class, see Figure 3A and 3B. The lubricating oil was
analysed using two independent temperature ramps of the GC×GC (methodologies outlined above);
one to achieve the best possible comprehensive separation of compounds in the oil (Figure 3A) and
the other using methodologies developed for analysis of the particulate phase components of engine
exhaust (Figure 3B), to ascertain where the compounds identified in the oil are present in the
particulate phase emissions filter. Figure 3B also illustrates the positioning of the SVOC measured in
the gas phase, that are observed in the particulate phase filter as well as the positioning for the PAH.
The grouping template that is illustrated in Figure 1 covers the SVOC range indicated in Figure 3B.


Using the signature mass fragment ions (Table 1) together with the calculated molecular mass,
specific compounds with the same carbon number were isolated, see Figure S9-A and S9-B. For





example, selecting the ion fragment $m/z$ 92 and 119 for monocyclic aromatics gives rise to the selected
ion chromatogram illustrated in Figure S9-A. This can be achieved using 70eV mass spectrometry
identifying a homologous series across a large carbon number range. However, selecting the
molecular mass for a specific carbon number allows the identification of all isomer sets in a region
of the chromatogram with that specific molecular mass, as shown in Figure S9B for $C_{22}$ monocyclic
aromatics ($m/z$ 302). A mass of 8511 ng of lubricating oil was injected into the GC×GC, of which
6356 (±1525) ng was quantified. This methodology was used to identify and quantify the following
homologous series: $C_{16}$–$C_{33}$ straight and branched chain alkanes, $C_{16}$–$C_{33}$ monocyclic alkanes, $C_{17}$–
$C_{33}$ bicyclic alkanes, $C_{17}$–$C_{33}$ tricyclic alkanes and $C_{16}$–$C_{33}$ monocyclic aromatics. These compound
groups represented approximately 91% of the total ion current (excluding siloxanes) and 75% of the
mass fraction. Adamantanes, diamantanes, pentacyclic and hexacyclic alkanes, steroids, steranes and
hopanes represented 5% of the total ion current, while the remaining 4% remained unidentified. These
compounds were not quantifiable using this methodology, as there were no standards available that
corresponded to these sections of the chromatography and could not be estimated as they are not
present in a homologous series. However, from previous literature, these compound classes are
thought to represent a small fraction of the mass (Worton et al., 2015).


Worton et al. (2015) exploited VUV photoionization mass spectrometry to characterize
comprehensively hydrocarbons in a standard reference crude oil sample. They reported a total mass
closure of 68 ± 22%, comprised of linear and branched alkanes (19%), 1-6 ring cycloalkanes (37%),
monoaromatics (6.8%) and PAH (4.7%). The mass fractions observed for linear and branched alkanes
in this study were 11% and 12%, respectively, which is in excellent agreement. There is also excellent





agreement with the mass attributed to bicyclic (2-ring) and tricyclic (3-ring) alkanes however, for
monocyclic alkanes the results presented here are a factor of 2 larger than Worton et al.[15] and 2.5
larger than Reddy et al. (2012). Both previous studies analysed similar crude oil samples associated
with the Deepwater Horizon disaster (McNutt et al., 2012), and would be expected to differ
appreciably from a lubricating oil. Furthermore, no PAH were observed in the lubricating oil in this
study, in agreement with Zielinska et al. (2004) but in contrast to Worton et al. (2015). We attribute
this difference to the varying crude oil origins and formulation processes involved in the production
of synthetic oil.

Previous work from this group identified a large number of isomeric species in base oil using 14eV
EI ionization energy mass spectrometry (Alam et al., 2016a). Although we were able to identify a
large number of compounds, there still existed a small amount of fragmentation at 14eV, particularly
for alkyl-methyl-, alkyl-dimethyl-, and alkyl-trimethyl-cyclohexanes. In this study the fragmentation
was significantly reduced for these compounds at 12eV (i.e. relative intensities of *m/z* 97, 111, 125
reduced by >50%) and completely eradicated (relative intensities of mass fragments reduced by >95%)
at 10eV, leaving the *m/z* 82 ion (for monocyclic alkanes) and the molecular ion. This demonstrates
the significant differences observed in fragmentation over small changes in lower ionisation EI
energies and may also account for slight discrepancies between studies (Worton et al., 2015; Isaacman
et al., 2012; Alam et al., 2016b).   Utilising these differences in fragmentation from using low
ionization energies (10 – 15eV) may provide more information in regards to the structure of many
isomeric compounds.

**3.4      Analysis of Diesel Engine Emissions (Particulate Phase)**





90% of the total ion current of the particulate phase filter was identified and attributed to a wide range
of classes. Of the total mass identified, 47 ($\pm$11)% was straight and branched chain alkanes, 20 ($\pm$4.8)%
monocyclic alkanes, 7.5 ($\pm$1.8)% bicyclic alkanes, <3 ($\pm$0.7)% tricyclic alkanes, 6 ($\pm$1.4)%
monocyclic aromatics, 7 ($\pm$1.7)% oxygenates, <1 ($\pm$0.2)% furanones, 4 ($\pm$1.0)% PAH and 2 ($\pm$0.5)%
fatty acid methyl esters (FAMES). Figure 4 illustrates the percentage mass contribution of
homologous series (including isomers) identified as a function of carbon number. Peak concentrations
of alkanes (cyclic and straight/branched chain) were observed between $C_{24} - C_{27}$ consistent with the
lubricating oil, while a small peak in concentration was also observed in the $C_{15} - C_{20}$ range,
consistent with the fuel and gas phase emissions. Oxygenated compounds were found to be present
in the $C_{11} - C_{22}$ range, suggesting that these compounds are combustion products. The concentration
of monocyclic aromatics was steady throughout the carbon number distribution ($C_{15} - C_{32}$), with a
small peak at $C_{25} - C_{27}$. The presence of PAH in the particulate phase suggests their formation via
diesel fuel combustion or unburnt fuel, owing to their absence in the lubricating oil. There are
numerous studies reporting the absence of PAH in unused lubricating oil and presence in used oil,
which suggests the absorption of blow-by exhaust containing fuel combustion associated PAH (Fujita
et al., 2006). FAMES were identified by their characteristic fragmentation at 70eV EI ionization and
with their characteristic ion (m/z 174) at low EI ionization (12 eV).


There have been few studies investigating the contribution of lubricating oil and fuel to the emitted
diesel POA, suggesting 20 to 80% influence from lubricating oil (Worton et al., 2014; Brandenberger
et al., 2005; Kleeman et al., 2008; Sonntag et al., 2012).    Most recently, it has been suggested that
$\geq$80% of the SVOC composition is dominated by branched cycloalkanes with one or more rings and





one or more branched alkyl side chain (Sakurai et al., 2003).[11] This is significantly larger than that
reported in this study (≥30 %), where the majority of the emissions are dominated by straight and
branched chain alkanes (47%) over a volatility range that also suggests a significant contribution from
the diesel fuel ($C_{11} - C_{20}$, see Figure 4). The diesel fuel and lubricating oil contained respectively 62%
and 47.5% straight and branched chain alkanes (summed), suggesting a larger possible contribution
of diesel fuel to the vapour phase engine emissions (which is dominated by straight and branched
chain alkanes). The contribution of unburned lubricating oil, however, most likely dominates the
SVOC emissions in the particulate phase, as shown in Figure 4.

**5.    CONCLUSION**
The SVOC content in diesel fuel, 5W30 synthetic lubricating oil and diesel exhaust emissions (both
in the gas and particulate phases) were characterized using TD-GC×GC-ToFMS. By exploiting the
mass spectrometric fingerprint of eluting compounds in highly structured and ordered chromatograms,
a methodology has been constructed in order to quantify the contributions of 'isomer sets' (i.e.
structural isomers in specific compound classes) to the overall composition of a sample. We found
that the ion current for identified homologous series exhibited very similar responses, illustrating that
quantitative calibrations derived from the *n*-alkane series could be used to estimate the concentrations
of isomeric aliphatic compounds with similar molecular weight. Using this methodology together
with a range of standards, and aggregating compound classes of similar functionality together (i.e. *n*-
alkanes, branched alkanes etc.), we present comprehensive characterization of diesel fuel, lubricating
oil and diesel exhaust emissions.






Furthermore, combining conventional 70eV EI ionization mass spectrometry with lower ionization
energy (10-14eV), allowed the identification of constitutional isomers of the same molecular weight
and compound class enabling a clear distinction between carbon number and functionality. By
utilising this innovative method, a number of finding were achieved: 1) a mass closure accounts for
*ca.* 90 % and 75 % for the analysis of diesel fuel and lubricating oil, respectively; 2) acyclic and
monocyclic alkanes were predominant in both the diesel fuel and synthetic lubricating oil (76% and
68%, respectively); 3) diesel exhaust emissions in the gas phase were extremely similar to the
composition of diesel fuel; 4) diesel exhaust emissions in the particulate phase were similar to the
composition of lubricating oil; 5) the presence of combustion products of diesel fuel (e.g. aromatics
and oxygenates) in the particulate phase indicates a contribution to POA.

Diesel exhaust hydrocarbons are a significant precursor of secondary organic aerosol (Dunmore et
al., 2015; |Gentner et al., 2012).  Diesel fuel and lubricant, contributors to diesel exhaust, contain
large numbers of isomers.  Separation into isomer sets is potentially a key step forward in
understanding the fates of these oil-derived materials in the environment (Lim & Ziemann, 2009;
Kroll & Seinfeld, 2008).  By utilizing GC×GC-ToFMS with soft ionisation, we enable the
identification of the composition of the UCM, characterising the chemical composition by carbon
number and compound class, and the possibility of branching structural information. Along with a
grouping methodology using CLIC expressions and unique compound fragmentation patterns, we
demonstrate the reliable quantitative integration of structural isomers. These methods exploit the
improved resolution and isomer separation capabilities of the advanced instrumentation and have
potential applications to the observations of petroleum degradation, and SOA formation and evolution.
This method can be extended to atmospheric measurements where there exist many oxygenates.




Although this adds chromatographic complexity, as co-elution can be a limitation, using carefully
constructed CLIC expressions and mass fragmentation patterns, various oxygenates can be identified
(*e.g.* 2-ketones (m/z 58), 3-ketones (m/z 72), carboxylic acids (m/z 60) *etc*.). It also has application
in any scientific field that routinely characterizes complex hydrocarbon mixtures, e.g., atmospheric
chemistry, microbial and chemical ecology, bioremediation, and aquatic pollution.


**ACKNOWLEDGEMENTS**
The authors would like to thank L. McGregor and N. Bukowski on valuable discussions on aspects
of GC×GC-ToFMS and assistance in CLIC qualifiers. Financial support from the European Research
Council (ERC-2012-AdG, Proposal No. 320821) for the FASTER project is gratefully acknowledged.
The assistance of Y. Al-Qahtani in collecting samples from the engine facility is also acknowledged.

**Author Contribution – needs to be completed**
MSA prepared the manuscript with contributions from ARM and RMH; RMH, MSA and SZR
designed the engine experiments; SZR, MSA and ZL carried out the engine experiments; MSA and
CS developed the GC×GC methodology and completed subsequent analyses. HX overlooked the
engine facility and RMH overlooked the entire project.

**Conflict of Interests**
The authors declare no competing financial interest.





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





**TABLE LEGEND:**

**Table 1.**    Hydrocarbons identified in diesel fuel, lubricating oil and diesel emissions (gas and
particulate phases) with their respective m/z fragment ions and percentage mass
contributions.


**FIGURE LEGENDS:**

**Figure 1.**    A contour plot (chromatogram) of diesel fuel separation. Peak height (intensity)
increases with warmth (blue to red) of the colour scale. Each region fenced by a coloured
polygon marks out the 2-dimensional chromatogram space in which are found isomers
of a particular compound type having a particular carbon number (*e.g.* $C_4$–substituted
monocyclic aromatics).

**Figure 2.**    A contour plot (chromatogram) displaying $C_6$-substituted monocyclic aromatics
identified by the CLIC expression. All $C_6$-substituted monocyclic aromatics are located
within the pink polygon displayed. 70eV (red peaks) and 12eV (blue peaks) mass spectra
corresponding to the peaks identified by the CLIC expressions in the SIC is shown for
6 different $C_6$-substituted monocyclic aromatics isomers. .

**Figure 3.**    A chromatogram of lubricating oil (5W30) **(A)** with labelled compound classes, using a
methodology specific for characterising the composition of lubricating oil; **(B)** using
methodology developed for characterising particulate phase emissions from diesel
engine exhaust. Polygons drawn around sections of the chromatograms indicate
compounds with different molecular masses within compound classes.

**Figure 4.**    Percentage mass contribution of the compounds identified in homologous series as a
function of carbon number in diesel exhaust particles.















**Table 1.** Hydrocarbons identified in diesel fuel, lubricating oil and diesel emissions (gas and particulate phases) with their respective m/z fragment ions and percentage mass contributions

| Compound Class | m/z (M$^+$)* | % Mass closure | | % Contribution to mass identified in emissions | |
|---|---|---|---|---|---|
| | | Diesel Fuel | Lubricating Oil | Gas Phase | Particulate Phase |
| **Total** | | **89.7** | **74.7** | **85.0 of TIC** | **75.0 of TIC** |
| n + i-Alkanes (C$_{11}$ – C$_{33}$) | 57 (C$_n$H$_{2n+2}$) | 62.2 | 23.0 | 39.9 | 47.3 |
| Monocyclic Alkanes (C$_{11}$ – C$_{33}$) | 82 (C$_n$H$_{2n}$) | 13.8 | 35.6 | 17.4 | 19.6 |
| Bicyclic Alkanes (C$_{11}$ – C$_{33}$) | 137 (C$_n$H$_{2n-2}$) | 5.0 | 9.2 | 9.7 | 7.5 |
| Tricyclic Alkanes (C$_{11}$ – C$_{33}$) | 191 (C$_n$H$_{2n-4}$) | <0.1 | 2.7 | <0.1 | 2.7 |
| Monocyclic Aromatics (C$_{11}$ – C$_{33}$) | 92, 119 (C$_n$H$_{2n-6}$) | 4.4 | 4.2 | 23.5 | 6.0 |
| Bicyclic Aromatics (C$_{11}$ – C$_{33}$) | 128, 141 | 0.8 | <0.1 | 2.0 | |
| Adamantanes | 135, 149, 163, 177 | | | | |
| Diamantanes | 187, 188, 201, 215, 229 | | | | |
| Pentacyclic Alkanes | 258, 272, 286 | | | | |
| Hexacyclic Alkanes | 298, 312 | | | | |
| Steroids | 239, 267 | | | | |
| Monoaromatic Steranes | 253 | | | | |
| Steranes | 217, 218 | | | | |
| Methyl Steranes | 217, 218, 231, 232 | | | | |
| 25-norhopanes | 177 | | | | |
| 28, 30-norhopanes | 163, 191 | | | | |
| Hopanes | 336 | | | | |
| PAHs | | 0.8 | | 0.6 | 4.0 |
| Biphenyls / Acenaphthenes | | 0.1 | | 0.1 | <0.1 |
| Tetralin / Indanes | 132, 145 | 2.5 | | 6.9 | |
| Oxygenates | | | | | 7.1 |
| Furanones | 84 | | | | 0.9 |
| FAMEs | 174 | | | | 2.0 |
| Miscellaneous Compounds | | | | | <3.0 |

*m/z ratios presented here are the main mass fragments present in the low ionisation energy mass spectra. CLIC expressions and 70eV mass spectra use more m/z fragments which were also used for qualification and quantification






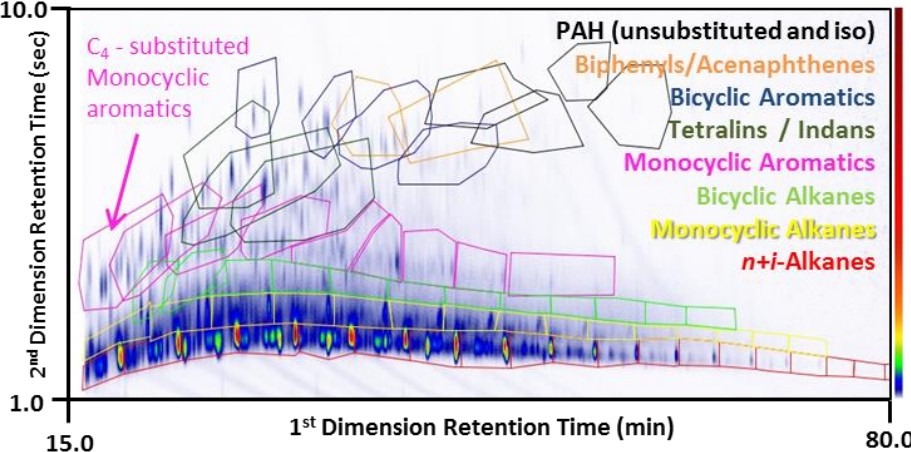

**Figure 1.** A contour plot (chromatogram) of diesel fuel separation. Peak height (intensity) increases with warmth (blue to red) of the colour scale. Each region fenced by a coloured polygon marks out the 2-dimensional chromatogram space in which are found isomers of a particular compound type having a particular carbon number (*e.g.* $C_4$–substituted monocyclic aromatics).





























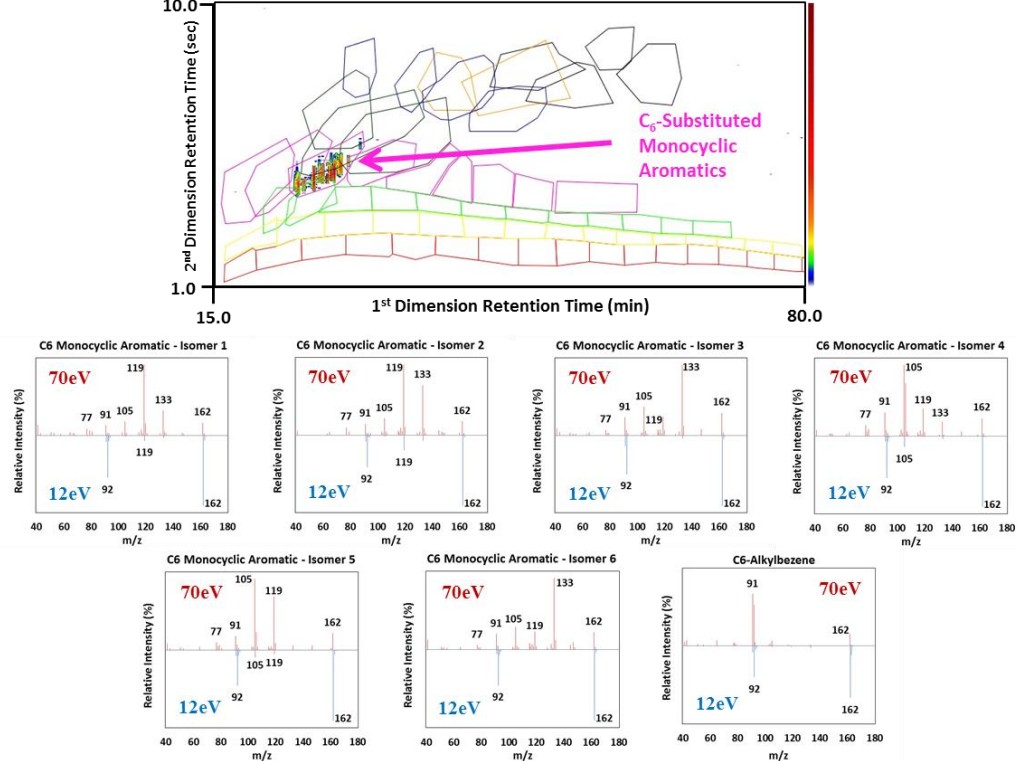

**Figure 2.** A contour plot (chromatogram) displaying $C_6$-substituted monocyclic aromatics identified by the CLIC expression. All $C_6$-substituted monocyclic aromatics are located within the pink polygon displayed. 70eV (red peaks) and 12eV (blue peaks) mass spectra corresponding to the peaks identified by the CLIC expressions in the SIC is shown for 6 different $C_6$-substituted monocyclic aromatics isomers.



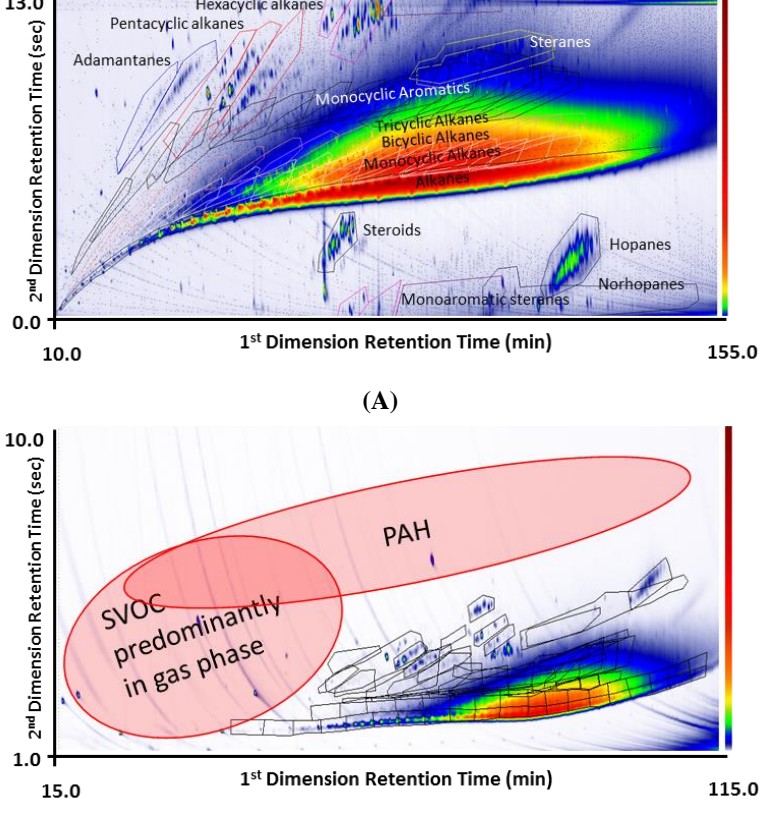

**(A)**

**(B)**

**Figure 3.** A chromatogram of lubricating oil (5W30) **(A)** with labelled compound classes, using a methodology specific for characterising the composition of lubricating oil; **(B)** using methodology developed for characterising particulate phase emissions from diesel engine exhaust. Polygons drawn around sections of the chromatograms indicate compounds with different molecular masses within compound classes.







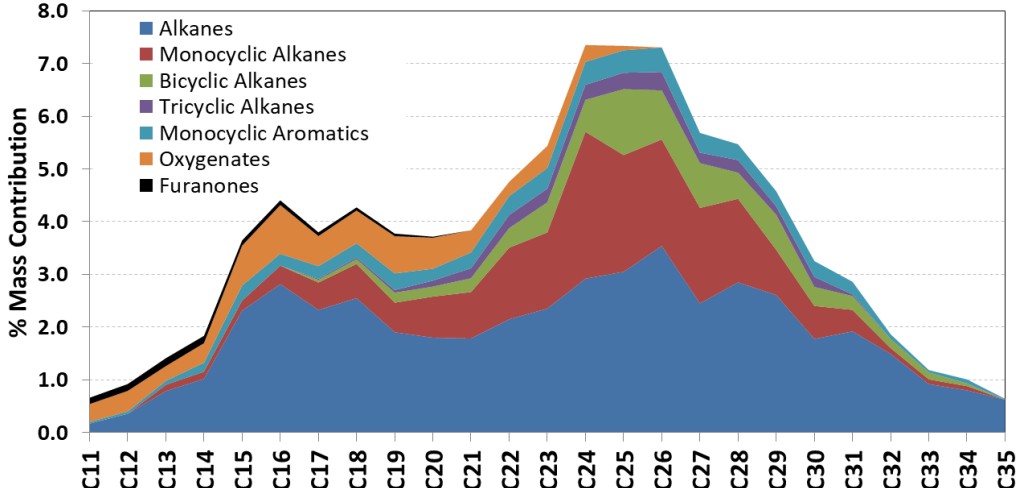

**Figure 4.** Percentage mass contribution of the compounds identified in homologous series as a function of carbon number in diesel exhaust particles.
