# Peer review of "Mapping and quantifying isomer sets of hydrocarbons (≥C₁₂) in diesel fuel, lubricating oil and diesel exhaust samples using GC×GC-ToFMS"

_Atmospheric Measurement Techniques, 2017_

## Referee Comment (RC1) · Anonymous Referee #2 · 20 Jan 2018

This article describes a novel GCxGC-TOF-MS approach for the characterization of carbonaceous material (oil and aerosol emissions). The research is novel and worth publishing although it is not clear why this necessarily belongs in an atmospheric journal as most of the work and discussion is on oil. The only, but serious weakness is in the writing itself. The text looks more like being part of an extensive report or a thesis than a scientific manuscript with constant referring to 3 other papers by the authors as well as abundant references to the SI. In its current form it is a painful read, even for people interested in the topic, and I find that it is not really able to stand on its own without the information in these other sources. I think the manuscript should be substantially cleaned up to be more readable and tell a consistent and concise story, with

the critical information if really not in the paper itself then in 1 single place. For the least, some clarifications are needed and this being an atmospheric measurement journal, it would be useful to emphasize the atmospheric relevance by a clearer comparison to what is known in the field.

Major points:

Please review what is really necessary here as method vs what can be in SI and vice versa and in regards to the other manuscripts. Some critical info is lacking as example: what mass resolution is used for the quantification? In no place the resolution of the TOF is mentioned and some spectra in SI are at unity resolution while others are at .1 Da resolution? This would matter.

The text should be focused more on the atmospheric community. This starts with the abstract where quantitative info is only given for oil and goes through the text. It is quite unsatisfying to just point out that the results are different from e.g. Gentner's work but then not attempt at all is being made to explain why? the same is true for other places of the manuscript where the text leaves it at: we see more/less than these other people. . . These differences should be addressed.

In that regards too, there are the highly cited papers by the Cass group (Rogge et al papers) that discuss for 1D GC extractable vs elutable, resolvable, identifiable for oils and diesel emissions. . . given that these papers are extremely known and cited, may be data could be compared? To illustrate the advance of this methodology over the 20 year old work that is cited hundreds of times.

The methodology of the quantification should be clarified in the present manuscript. example~ L254-2" Therefore, out of the 8026 ng that was injected into the GC×GC, a mass of approximately 7200 ($\pm$ 1728) ng was accounted for." It is unclear here. . . how the 8026 ng was determined? and why does it not have an uncertainty? Similarly how was the uncertainty for the 7200 obtained?

[Figure]

Please clarify how some homologous series were quantified for which you do not list standards (hopanes and steranes etc)?

I am deeply confused also on the mass balance how any non organic/hydrocarbon species are handled? Especially synthetic motor oil could have a substantial fraction Si or other non HCO materials. Related also you refer to synthetic oil and contribution form crude. This is confusing as synthetic oil could be completely synthetic i.e. nothing in there has ever seen crude.

Minor points:

The text needs some attention to detail. In the experimental section, please use a consistent form to refer to manufacturer, location etc.. also check for typos, USA not US etc Do states need to be included for USA?

I am not a native speaker but the English clearly seems off at times e.g. 187-190, two sub sentences starting with while.. reads really odd?

References cite in chronological and antichronological order in the text not random

Details: L129: Ref: EU. . . please write as citation

L153: "6"? number reference?

L253 I am not sure it is appropriate to label column bleed as contaminant plus there is no evidence that this is only column bleed. Siloxanes might well be present in synthetic oil or even ambient air.

L322.. this is experimental and does not belong in the results section

SI section:

Page 4: formatting issue

Page 7: Liang et al 2017 in prep?

Figure S4-1: legend X axis? What is 1/MW (C24-C12 Molar Quantity)? Explain clearer

and make legend text consistent with actual legend

S4-2 left panel idem to S4-1 plus straight unreadable (font size)

S4-3 make labeling consistent with other figures

All of the double MS plots, please make them look like the ones in the actual paper or I suggest that on one panel at least the eV are indicated as for MS community this way of showing spectra is typically how + and – ion spectra, so could be confusing

The model spectra, are they corrected for background ? how were they obtained? some have many peaks (PAH)?

S6 what is really the use of this if not even the position of one relative to the other are shown in fig 6

Please even in SI use a uniform formatting of references

---

## Referee Comment (RC2) · Anonymous Referee #1 · 4 Feb 2018

In this work, the authors Alam et al. present a novel method to characterize the composition of organic mixtures, especially those from fossil fuel combustion sources. This is an important topic for atmospheric sciences, as many of these mixtures can react in the atmosphere and form oxygenated compounds and organic aerosol. This work uses a similar approach as work by Zimmerman et al. and Goldstein et al., and combines low energy ionization with multidimensional chromatography to separate and identify aliphatic and aromatic isomers. This work is of clear interest to the field, and is within the scope of AMT. Overall, the work here is quite solid. I suggest a few more discussion points that would advance the understanding of the technique, so that the technique is more likely to be adopted by researchers in this field.

[Figure]

1. Ionization temperature plays an important role in the fragmentation of the molecule. The thermal energy can lead to substantial fragmentation, e.g. Isaacman et al. show that lowering ion source temperature from 300C to 150C reduces fragmentation temperature significantly. Amirav et al. (2008) uses a supersonic molecular beam to achieve cold electron impact ionization, and was able to reduce fragmentation substantially. Have the authors tried to lower ionization source temperature?

2. Is isomer-level differentiation/quantification necessary? There is some debate whether or not the isomers are different enough to warrant detailed characterization. I suggest the authors look at Tkacik et al. (2012) and Goodman-Rendall et al. (2016). In other words, it might be sufficient for process level understanding of hydrocarbon composition (e.g. volatility basis set approach or 2-product Odum model) to predict oxidation characteristics and volatility changes. I think this is an important point: the authors will first need to identify the need to perform this level of chemical characterization before they will be able to convince the field that this technique is useful.

3. Can the authors explain why the total ion current is so similar between different compound types? Ion sensitivity is a strong function of electron absorption cross section, and at lower energies, there is significant differences in cross section. Aromatic compounds, for example, would absorb much more effectively than aliphatic compounds, and cyclic compounds (six- and five-carbon ring) would also absorb more than cyclic compounds. I understand that the authors are just showing their results, but they seem a little counter-intuitive to me.

4. While this technique would be ideal for investigating fossil fuel sources (e.g. motor vehicles, oil extraction activities), how would this work for atmospheric mixture where there will be many other types of compounds and sources? In urban areas, the HOA factor from aerosol mass spectrometer studies accounts for around 20% of total OA, so the majority will be oxygenated. What happens when an urban aerosol sample is analyzed using this technique? Will the "contamination" by oxygenated compounds limit the ability to measure the hydrocarbons?

5. Given that this is an atmospheric journal, the writing should reflect the focus on atmospheric applications. For example, the discussion about other environmental mixtures in the conclusions may not be necessary and seem to detract from the main point.

Minor comments:

1. Samples from the diesel exhaust are obtained from extraction using dichloromethane. What is the extraction efficiency for oxygenated compounds using DCM as a solvent?

2. Lines 231-233: How reproducible are the retention times in both dimensions? In other words, are the polygons drawn to separate different compound types consistent from one run to another, one column set to another (assuming the same type of columns), one day to another?

3. Line 310: Carboxylic acids are very difficult to be detected by GC/MS without prior derivatization. If they are detected, their quantities might be highly uncertain. It might be difficult to say whether they are truly minor, or it is just a limitation of GC/MS.

4. Line 274: aren't saturated cyclic alkanes also alkanes?

References:

Goodman-Rendall, K.A., Zhuang, Y. R., Amirav, A., Chan, A. W. H., 2016. Resolving detailed molecular structures in complex organic mixtures and modeling their secondary organic aerosol formation. Atmos. Env. 128, 276-285.

Tkacik, D.S., Presto, A.A., Donahue, N.M., Robinson, A.L., 2012. Secondary organic aerosol formation from intermediate-volatility organic compounds: cyclic, linear, and branched alkanes. Environ. Sci. Technol. 46, 8773-8781.

---

## Author Comment (AC1) · 2 Mar 2018

In this work, the authors Alam et al. present a novel method to characterize the composition of organic mixtures, especially those from fossil fuel combustion sources. This is an important topic for atmospheric sciences, as many of these mixtures can react in the atmosphere and form oxygenated compounds and organic aerosol. This work uses a similar approach as work by Zimmerman et al. and Goldstein et al., and combines low energy ionization with multidimensional chromatography to separate and identify aliphatic and aromatic isomers. This work is of clear interest to the field, and is within the scope of AMT. Overall, the work here is quite solid. I suggest a few more discussion points that would advance the understanding of the technique, so that the technique is more likely to be adopted by researchers in this field.

**RESPONSE**

*We thank the anonymous reviewer's comments and have addressed the comments raised below.*

1. Ionization temperature plays an important role in the fragmentation of the molecule. The thermal energy can lead to substantial fragmentation, e.g. Isaacman et al. show that lowering ion source temperature from 300C to 150C reduces fragmentation temperature significantly. Amirav et al. (2008) uses a supersonic molecular beam to achieve cold electron impact ionization, and was able to reduce fragmentation substantially. Have the authors tried to lower ionization source temperature?

**RESPONSE**

*We have not lowered the ionisation source temperature to reduce the fragmentation in this study. Rather we have focused on reducing the ionisation energy only to reduce fragmentation. Using the select-eV BenchTOF, the user can amend the ionisation between 10eV and 70eV and we have assessed the mass fragmentation patterns between 10-15eV and 70eV ionisation energies, without changing the ion source temperature. The transfer line and ion source temperatures used in this study are the recommended temperatures by the manufacturer (Markes International). We expect the fragmentation to reduce and to retain a larger fraction of the molecular parent ion if we were to reduce the ion source temperature, like that reported by Isaacman et al. However, this has not been incorporated into this study as it is independent of ion source temperature.*

*A sentence has been added to the end of the conclusion. See lines 540 – 547.*

2. Is isomer-level differentiation/quantification necessary? There is some debate whether or not the isomers are different enough to warrant detailed characterization. I suggest the authors look at Tkacik et al. (2012) and Goodman-Rendall et al. (2016). In other words, it might be sufficient for process level understanding of hydrocarbon composition (e.g. volatility basis set approach or 2-product Odum model) to predict oxidation characteristics and volatility changes. I think this is an important point: the authors will first need to identify the need to perform this level of chemical characterization before they will be able to convince the field that this technique is useful.

**RESPONSE**

*Volatility is not the only important characteristic of a VOC in determining its fate in the environment or its environmental impact. We have included an additional few sentences in the introduction to include these important points raised by the reviewer. See lines 72 – 76 and 115 to 125.*

3. Can the authors explain why the total ion current is so similar between different compound types? Ion sensitivity is a strong function of electron absorption cross section, and at lower energies, there is significant differences in cross section. Aromatic compounds, for example, would absorb much more effectively than aliphatic compounds, and cyclic compounds (six- and five-carbon ring) would also absorb more than cyclic compounds. I understand that the authors are just showing their results, but they seem a little counter-intuitive to me.

**RESPONSE**

*Chromatograms shown in Figures 1, 3 and S5 and S7A in this study are those performed with 70eV ionisation mass spectrometry, which we use for quantitative analysis. The 12eV spectra we use qualitatively (see Figure 2 and S7B). This may explain to the reviewer why the total ion current is so similar between the different compound types – because of the relatively large energy used for fragmentation.*

*A sentence has been added to the manuscript to explain this. See lines 237 – 238.*
*All figures of chromatograms within the manuscript and supplementary information have included their respective ionisation energies of the mass spectrometer in their legends.*

4. While this technique would be ideal for investigating fossil fuel sources (e.g. motor vehicles, oil extraction activities), how would this work for atmospheric mixture where there will be many other types of compounds and sources? In urban areas, the HOA factor from aerosol mass spectrometer studies accounts for around 20% of total OA, so the majority will be oxygenated. What happens when an urban aerosol sample is analyzed using this technique? Will the "contamination" by oxygenated compounds limit the ability to measure the hydrocarbons?

**RESPONSE**

*We have used this technique on atmospheric samples and have found that although there are an increased number of oxygenated species present, we can still adequately identify and quantify the hydrocarbons that are measured in this study. In the gas phase the majority of the aliphatic oxygenated compounds are present in specific bands above the monocyclic alkanes and below the monocyclic aromatics which can be identified in the atmospheric samples using additional CLIC expressions and fragmentation patterns (as outlined in the conclusion lines 537 – 540).*
*A section has been incorporated into the conclusion to expand on this point made by the reviewer (see Conclusion, line 532 - 541)*

5. Given that this is an atmospheric journal, the writing should reflect the focus on atmospheric applications. For example, the discussion about other environmental mixtures in the conclusions may not be necessary and seem to detract from the main point.

**RESPONSE**

*This has been removed. We believe that this manuscript is now more focused on the atmospheric community with the further modifications that we have made. We have also*

*modified the paper title to: "Mapping and quantifying isomer sets of hydrocarbons ($\geq C_{12}$) in diesel exhaust, lubricating oil and diesel fuel samples using GC×GC-ToFMS"*

*See lines 28 – 29, 44 – 45, 72 – 76, 115 – 125, 146 – 158 and 532 – 547*

**Minor comments:**

1. Samples from the diesel exhaust are obtained from extraction using dichloromethane. What is the extraction efficiency for oxygenated compounds using DCM as a solvent?

**RESPONSE**

*We have spiked a known concentration of an oxygenated standard (2-heptadecanone) onto a filter, extracted using DCM and analysed using the GCxGC and found that the extraction efficiency for that oxygenated compound was >92%. This gives us confidence that the extraction with DCM can obtain oxygenated compounds. The reason why we chose DCM as a solvent was to extract as many of the compounds present on the filter as possible, over the polar and non-polar range. We have performed multiple analyses with hexane, DCM and methanol as extraction solvents, and found that DCM performed the best in extracting the majority of the polar/non-polar compounds.*

*We have included a sentence in the supplementary information in Section S1.*
*See lines 38 – 40.*

2. Lines 231-233: How reproducible are the retention times in both dimensions? In other words, are the polygons drawn to separate different compound types consistent from one run to another, one column set to another (assuming the same type of columns), one day to another?

**RESPONSE**

*Yes. The retention times are reproducible in both dimensions in separate runs. The polygons are all linked together in a large template that can also be linked to the internal standards. This means that it is easy to align the template (all polygons) in the event of slight shift in retention times. These shifts, however, only occur when the columns are changed or if there has been some major maintenance performed on the instrument.*

*We have included a few sentences to the manuscript to explain this. See lines 284 – 287.*

3. Line 310: Carboxylic acids are very difficult to be detected by GC/MS without prior derivatization. If they are detected, their quantities might be highly uncertain. It might be difficult to say whether they are truly minor, or it is just a limitation of GC/MS.

**RESPONSE**

*The reviewer raises a good point and this has been incorporated into the manuscript.*
*See lines 380 – 381.*

4. Line 274: aren't saturated cyclic alkanes also alkanes?

**RESPONSE**

*Yes, but we refer to straight chain saturated linear alkanes as "linear alkanes" and saturated cyclic alkanes as "cyclic alkanes".*
*The manuscript remains unchanged.*

**References:**

Goodman-Rendall, K.A., Zhuang, Y. R., Amirav, A., Chan, A. W. H., 2016. Resolving detailed molecular structures in complex organic mixtures and modeling their secondary organic aerosol formation. Atmos. Env. 128, 276-285.

Tkacik, D.S., Presto, A.A., Donahue, N.M., Robinson, A.L., 2012. Secondary organic aerosol formation from intermediate-volatility organic compounds: cyclic, linear, and branched alkanes. Environ. Sci. Technol. 46, 8773-8781.

This article describes a novel GCxGC-TOF-MS approach for the characterization of carbonaceous material (oil and aerosol emissions). The research is novel and worth publishing although it is not clear why this necessarily belongs in an atmospheric journal as most of the work and discussion is on oil. The only, but serious weakness is in the writing itself. The text looks more like being part of an extensive report or a thesis than a scientific manuscript with constant referring to 3 other papers by the authors as well as abundant references to the SI. In its current form it is a painful read, even for people interested in the topic, and I find that it is not really able to stand on its own without the information in these other sources. I think the manuscript should be substantially cleaned up to be more readable and tell a consistent and concise story, with the critical information if really not in the paper itself then in 1 single place. For the least, some clarifications are needed and this being an atmospheric measurement journal, it would be useful to emphasize the atmospheric relevance by a clearer comparison to what is known in the field.

**RESPONSE**

*We thank the anonymous reviewer for their comments and have addressed their comments below.*

*We believe that this study belongs to an atmospheric journal as it presents data not only regarding oil and fuel, but also includes detailed analyses of gas and particulate phase diesel emissions; which are emitted into the atmosphere and can contribute to secondary organic aerosol. The technique of GCxGC-ToFMS has been adopted by the atmospheric science community in the past (the groups of Goldstein (Berkeley) and Lewis (York)), but has not yet exploited the use of variable or low ionisation energy mass spectrometry presented in this study. The work by Goldstein et al., and Zimmerman et al., have exploited photoionisation VUV for low ionisation MS; and this study presents the application of variable ionisation EI mass spectrometry to map and quantify isomer sets of hydrocarbons. This technique can be applied to complex matrices, particularly diesel emissions and atmospheric samples. For these reasons, we believe that this technique can be adopted by many researchers in the field, as anonymous reviewer 1 has already suggested.*

*The reviewer points out that the text refers to 3 other papers and to the SI substantially. We have addressed this by including some concise information within the manuscript so that the reader no longer has to refer to the other papers and/or the SI (unless they are interested in further information).*

*Major changes within the manuscript include:*

1) *The second paragraph in the Experimental section (2.1 sampling) has been edited to include information regarding the engine set up and sampling procedure. This*

*eliminates the referral to Alam et al. (2016c). What was formerly the second paragraph has been moved to the front of this section and is now paragraph one.*

2) *Section S3 in Supplementary Information has been modified and moved to the manuscript under a new additional Section 2.5 "Quantification of compounds with no authentic standards". Section S3 in the SI now only contains the uncertainty calculation for this method.*

3) *A sentence referring to the potential of scaling ion current to molar quantity (lines 302-304) and its corresponding section in the Supplementary Information (Section S4) has been removed.*

4) *Supplementary Section S6 has also been removed, as suggested by the reviewer below.*

5) *We have modified the paper title to: "Mapping and quantifying isomer sets of hydrocarbons ($\geq C_{12}$) in diesel exhaust, lubricating oil and diesel fuel samples using GC×GC-ToFMS"*

**Major points:**

Please review what is really necessary here as method vs what can be in SI and vice versa and in regards to the other manuscripts. Some critical info is lacking as example: what mass resolution is used for the quantification? In no place the resolution of the TOF is mentioned and some spectra in SI are at unity resolution while others are at .1 Da resolution? This would matter.

**RESPONSE**

*We have reviewed what should be in the manuscript and the SI concisely. Section S3 in Supplementary Information has been modified and moved to the manuscript under a new additional Section 2.5 "Quantification of compounds with no authentic standards". Section S3 in the SI now only contains the uncertainty calculation for this method. Also Sections S4 and S6 have been removed from the SI also.*

*The mass resolution used for quantification has been added to the manuscript together with the resolution of the TOF. We use a nominal resolution of unity Da for quantification. See lines 194 to 197.*

The text should be focused more on the atmospheric community. This starts with the abstract where quantitative info is only given for oil and goes through the text. It is quite unsatisfying to just point out that the results are different from e.g. Gentner's work but then not attempt at all is being made to explain why? the same is true for other places of the manuscript where the text leaves it at: we see more/less than these other people: : : These differences should be addressed.

**RESPONSE**

*The abstract has been amended to include quantitative information for gas and particulate phase emissions. We have also included additional text in the conclusion to address the use of this technique with atmospheric samples and oxygenated compound. We believe that this manuscript is focused on the atmospheric community with the modifications that we have made. We have also modified the paper title to: "Mapping and quantifying isomer sets of hydrocarbons ($\geq C_{12}$) in diesel exhaust, lubricating oil and diesel fuel samples using GC×GC-ToFMS"*

*See lines 28 – 29, 44 – 45, 72 – 76, 115 – 125, 146 – 158 and 532 – 547*

*The differences/similarities that we observe for diesel fuel with Isaacman et al. (2012) are attributable to the different fuel formulations and/or fuel source, as opposed to analytical methods. This is written in the manuscript (lines 342 – 344).*

*Line 374 has been deleted "in contrast with Worton et al. (2014)". This statement can be misleading as Worton et al. do not measure constituents in the gas phase and only focus on the particulate phase. In this section we state that we think it is unlikely for there to be a contribution from lubricating oil to the gas phase emissions.*

*For the analysis of lubricating oil we attribute the differences seen between the work of Worton et al. and ours to be the varying crude oil origins and formulation processes involved in the production of synthetic oil. This is stated in lines 442 – 447.*

In that regards too, there are the highly cited papers by the Cass group (Rogge et al papers) that discuss for 1D GC extractable vs elutable, resolvable, identifiable for oils and diesel emissions: : : given that these papers are extremely known and cited, may be data could be compared? To illustrate the advance of this methodology over the 20 year old work that is cited hundreds of times.

**RESPONSE**

*We have included the highly cited paper of Rogge et al. (1993) in the introduction of the manuscript but not compared the results in the discussion section as these studies were conducted 25 years ago where only 10-15% of the organics were resolved using GC/MS. Furthermore, the sulphur content in the diesel fuel would have been significantly larger than that of the diesel fuel today and so emissions from diesel exhaust would not be directly comparable. See Jones et al. (2012) for more information. (Jones et al., A large reduction in airborne particle number concentrations at the time of the introduction of "sulphur free" diesel and the London Low Emission Zone, Atmospheric Environment 50, 129-138). See lines 106 – 108.*

The methodology of the quantification should be clarified in the present manuscript. example_ L254-2" Therefore, out of the 8026 ng that was injected into the GC_GC, a mass of approximately 7200 (_ 1728) ng was accounted for." It is unclear here: : : how the 8026 ng was determined? and why does it not have an uncertainty? Similarly how was the uncertainty for the 7200 obtained?

**RESPONSE**

*The value of 8026 ng of diesel fuel was the calculated mass that was injected into the GC. A known volume of diesel fuel was weighed and diluted, followed by injection into the GC. The uncertainty associated to 8026 is the combined uncertainty of the apparatus that was used to calculate the mass (i.e. the combined uncertainties of the balance used to weigh the diesel fuel, the pipette used to measure the volume of the fuel and the volumetric flasks used to dilute the fuel; which is 0.3%). This has been added into the manuscript. See line 319-321.*

*The uncertainty of the mass that was accounted for using the technique is 24% which is the estimated uncertainty of the technique and is discussed in Section 2.5 in the manuscript and explained in detail in the SI Section S3.*

Please clarify how some homologous series were quantified for which you do not list standards (hopanes and steranes etc)?

**RESPONSE**

*Section S3 in Supplementary Information has been modified and moved to the manuscript under a new additional Section 2.5 "Quantification of compounds with no authentic standards". Section S3 in the SI now only contains the uncertainty calculation for this method.*

*We do not quantify hopanes and steranes as outlined in the manuscript. See lines 424 to 430 which states: "Adamantanes, diamantanes, pentacyclic and hexacyclic alkanes, steroids, steranes and hopanes represented 5% of the total ion current, while the remaining 4% remained unidentified. These compounds were not quantifiable using this methodology, as there were no standards available that corresponded to these sections of the chromatography and could not be estimated as they are not present in a homologous series. However, from previous literature, these compound classes are thought to represent a small fraction of the mass (Worton et al., 2015)."*

I am deeply confused also on the mass balance how any non organic/hydrocarbon species are handled? Especially synthetic motor oil could have a substantial fraction Si or other non HCO materials. Related also you refer to synthetic oil and contribution form crude. This is confusing as synthetic oil could be completely synthetic i.e. nothing in there has ever seen crude.

**RESPONSE**
*We have not taken into account any non-organic/hydrocarbon species but according to the data presented in this study the fraction of Si or other non HCO materials would be small.*

*Synthetic lubricating oil can be manufactured using chemically modified petroleum components rather than whole crude oil. This means that it has seen a fraction of crude oil.*

*A sentence has been added to the manuscript to include this point. See lines 430 – 432.*

**Minor points:**
The text needs some attention to detail. In the experimental section, please use a consistent form to refer to manufacturer, location etc.. also check for typos, USA not US etc Do states need to be included for USA?

**RESPONSE**
*The text has been amended to refer to manufacturer, location - city, location – country.*
*See line 164*
*States within the USA have been removed.*
*See line 188*
*Addition of "A" for USA.*
*See line 200*

I am not a native speaker but the English clearly seems off at times e.g. 187-190, two sub sentences starting with while.. reads really odd?

**RESPONSE**
*Replaced "while" in the first sub sentence to "at the same time as.."*
*See line 234 – 235.*

References cite in chronological and antichronological order in the text not random

**RESPONSE**
*Only three references that have been cited from the same author in the same year have all been cited in chronological order in the text. For example: Alam et al. (2016a) has been cited*

*in line 83, Alam et al. (2016b) has been cited in line 137 and Alam et al. (2016c) has been cited in line 158. We have checked the references thoroughly and have not identified any other issues.*

*The text remains unchanged.*

Details: L129: Ref: EU: : : please write as citation

**RESPONSE**

*Amended.*

*See line 167 – 168, reference added also, see line 617 – 619*

L153: "6"? number reference?

**RESPONSE**

*Replaced with Alam et al. (2016a)*

*See line 199*

L253 I am not sure it is appropriate to label column bleed as contaminant plus there is no evidence that this is only column bleed. Siloxanes might well be present in synthetic oil or even ambient air.

**RESPONSE**

*We have performed extensive analyses on ambient air samples, lubricating oils, diesel fuel and associated field and lab blanks. In all of these analyses we have observed these siloxanes, so we are very confident that this is column bleed.*

*The manuscript remains unchanged.*

L322.. this is experimental and does not belong in the results section

**RESPONSE**

*We have removed a portion of this section that is reproducing what is already in the experimental section (section 2.4). Lines 388 – 394 have been removed.*

**SI section:**

Page 4: formatting issue

**RESPONSE**

*This is not a formatting issue. This is the code that is used for the CLIC expressions that have been discussed in the manuscript (section 2.4). These are code that can be extracted and used as is if this technique was to be utilised.*

Page 7: Liang et al 2017 in prep?

**RESPONSE**

*This has been removed*

Figure S4-1: legend X axis? What is 1/MW (C24-C12 Molar Quantity)? Explain clearer and make legend text consistent with actual legend
S4-2 left panel idem to S4-1 plus straight unreadable (font size)
S4-3 make labeling consistent with other figures

**RESPONSE**

*This section has been removed altogether to keep the manuscript and supplementary information concise.*

All of the double MS plots, please make them look like the ones in the actual paper or I suggest that on one panel at least the eV are indicated as for MS community this way of showing spectra is typically how + and – ion spectra, so could be confusing

**RESPONSE**

*This has been amended and the eV has been labelled on each panel*
*See SI Figures S4-1 to S4-3 and Figure S6-1 to S6-7.*

The model spectra, are they corrected for background ? how were they obtained? some have many peaks (PAH)?

**RESPONSE**

*These are not modelled spectra. These mass spectra are experimentally determined at 70eV and 12eV like explained in the text in Section S5.*
*The text has been amended to add the words "Experimentally determined..." See line 224 in SI.*

*A mass spectrum is determined by selecting a peak of interest in the GC chromatogram. That peak will have its own mass spectrum and can then be compared to the NIST library for positive identification. This is the principle of operation of the GCxGC-TOFMS. If there is co-elution of peaks then the mass spectrum of those peaks would have been merged and will need to be deconvoluted using the GC Image software.*

*The reason that some of the mass spectra in Section S4 have multiple peaks is because of the extensive fragmentation at 70eV, which is discussed in detail throughout the manuscript.*
*The PAH have many peaks present at 70eV which is in line with the NIST 70eV library. For example, the 70eV MS for phenanthrene (see Figure S4-3 below, top right panel) possessed the mass fragments m/z 76, 89 and 152, corresponding to $C_6H_4$, $C_7H_5$ and $C_{12}H_8$ fragments, respectively. This mass spectrum is identical to that of phenanthrene in the NIST library at 70eV and is confirmed by the 12eV mass spectrum which only possesses the molecular mass 178.*

[Figure]

*Figure S4-3.* *Representative mass spectra of four PAH. Top graphs with red peaks are 70eV ionisation mass spectra, while bottom graphs with blue peaks are 12eV spectra.*

S6 what is really the use of this if not even the position of one relative to the other are shown in fig 6
**RESPONSE**
*This has been removed from the SI*

Please even in SI use a uniform formatting of references
**RESPONSE**
*Amended*

[revised manuscript text omitted]

**CONTENTS**

**Section S1: Chromatography Methodology**

Exposed adsorption tubes were spiked with 1 ng of deuterated internal standard mix for quantification, and desorbed onto the cold trap at 350 °C for 15 min (trap held at 20 °C). The trap was then purged onto the column in a split ratio of 102:1 at 350 °C and held for 4 min. An initial temperature of 90 °C of the primary oven was held for 2 min and then increased at 2 °C min$^{-1}$ to 240 °C, followed by 3 °C min$^{-1}$ to 310 °C and held for 5 min. The initial temperature of the secondary oven of 40 °C was held for 2 min, and then increased to 250 °C by 3 °C min$^{-1}$, followed by an increase of 1.5 °C min$^{-1}$ to 315 °C and held for 5 min. The modulation period was 10 s. The complete run time was approximately 105 min.

Diluted lubricating oil samples and particulate phase filters were spiked with 50 µL of 1 ng µL$^{-1}$ deuterated internal standard mix for quantification. Filters were immersed in dichloromethane (DCM) and ultrasonicated at 20 °C for 20 min. DCM was chosen as an extraction solvent as this was found to have a greater extraction efficiency over hexane and methanol in extracting the large range of polar and non-polar compounds of interest. The extract was then concentrated to 50 µL under a gentle flow of N$_2$ for analysis on the GC×GC-ToF-MS. 1 µL of the extracted sample was injected in a split ratio 100:1 at 300 °C. An initial temperature of 120 °C of the primary oven was held for 2 min and then increased at 2 °C min$^{-1}$ to 210 °C, followed by 1.5 °C min$^{-1}$ to 325 °C and held for 5 min. The initial temperature of the secondary oven of 120 °C was held for 2 min, and then increased to 200 °C by 3 °C min$^{-1}$, followed by an increase of 2 °C min$^{-1}$ to 300 °C and a final increase of 1 °C min$^{-1}$ to 330 °C, ensuring all species pass through the column efficiently. The modulation period was 13 s.

1 μL of diluted (1:1000) lubricating oil was injected in a split ratio 100:1 at 300 °C. An initial temperature of the primary and secondary ovens were kept the same (175 °C) held for 5 min. The primary oven temperature was increased by 1°C min$^{-1}$ to 325°C, while the secondary oven temperature was increased by 1°C min$^{-1}$ to 330°C. A modulation time of 8 s was used, while a total run time of each sample was 120 min. The transfer line and ion source temperatures were 325 °C

and 320 °C, respectively and were kept consistent for all sample analyses. Helium was used as the carrier gas at a constant flow rate of 1 mL min$^{-1}$.

**Section S2: Computer Language for Identification of Chemicals (CLIC qualifiers)**

Language functions: See Reference 2 (below) for more details on selection language functions.

*Ordinal* – Returns the ordinal position of the indicated channel (m/z in a mass spectrum) in the intensity-ordered multi-channel array of the current object (blob)

*Retention* – Returns the retention time of the current object (blob) with respect to the chromatographic column indicated by the dimension parameter. Retention time for dimension 1 is expressed in minutes and dimension 2 is expressed in seconds.

*Relative* – Returns the intensity value of the indicated channel (m/z in a mass spectrum) in the multi-channel intensity array of the current object (blob) as a relative percentage of the largest intensity value of the array.

See Reichenbach et al. (2005) for more details on selection language functions.

Some examples of CLIC qualifiers are shown below:

C2-Alkyl Benzenes
=(ordinal(91)=1)&(ordinal(106)=2);

C3-Alkyl Benzenes
=((ordinal(91)=1)&(ordinal(120)<=3)&(retention(1)<30))|((ordinal(105)=1)&(ordinal(120)<=3)&(r
etention(1)<30));

C4-Alkyl
Benzenes=((ordinal(91)=1)&(ordinal(134)<=3))|((ordinal(105)=1)&(ordinal(134)<=3)&(retention(
1)<80))|((ordinal(119)=1)&(ordinal(134)<=3));

C5-Alkyl
Benzenes=((ordinal(92)=1)&(ordinal(148)<=3))|((ordinal(105)=1)&(ordinal(148)<=4))|((ordinal(11
9)=1)&(ordinal(148)<=4))|((ordinal(133)=1)&(ordinal(148)<=3));

C6-Alkyl
Benzenes=((ordinal(92)=1)&(ordinal(162)<=4))|((ordinal(105)=1)&(ordinal(162)<=5))|((ordinal(11
9)=1)&(ordinal(162)<=5))|((ordinal(133)=1)&(ordinal(162)<=5))|((ordinal(147)<=2)&(ordinal(162
)<=5)&(retention(2)<6))|((ordinal(106)=1)&(ordinal(162)<=4))|((ordinal(91)=1)&(ordinal(162)<=4
));

C7-Alkyl
Benzenes=((ordinal(105)=1)&(ordinal(176)<=4))|((ordinal(133)=1)&(ordinal(176)<=4))|((ordinal(1
61)=1)&(ordinal(176)<=4)&(Retention(2)<5.5))|((ordinal(91)=1)&(ordinal(176)<=3))|((ordinal(147
)=1)&(ordinal(119)<=4)&(ordinal(176)<=4))|((ordinal(106)=1)&(ordinal(176)<=4))|((ordinal(119)
=1)&(ordinal(176)<=5));

C1-Alkyl NAP=(ordinal(141)<=3)&(ordinal(115)<=3)&(ordinal(142)<=3);

C2-Alkyl
NAP=((ordinal(141)=1)&(ordinal(156)=2)&(ordinal(115)=3)&(relative(141)>90)&(relative(156)>5
0))|((ordinal(156)=1)&(ordinal(141)=2)&(ordinal(155)=3)&(relative(156)>90)&(relative(141)>50))
;

C3-Alkyl NAP
=((ordinal(155)=1)&(ordinal(170)=2)&(Relative(155)>80)&(Relative(170)>25))|((ordinal(170)=1)
&(ordinal(155)=2)&(Relative(170)>80)&(Relative(155)>60))|((ordinal(141)=1)&(ordinal(170)=2)
&(ordinal(115)=3)&(Relative(141)>80)&(Relative(170)>25)&(Relative(115)>15));

C4-Alkyl
NAP=((ordinal(155)=1)&(ordinal(184)=2)&(Relative(155)>80)&(Relative(184)>15))|((ordinal(141
)=1)&(ordinal(184)=2)&(Relative(141)>80)&(Relative(184)>15))|((ordinal(169)=1)&(ordinal(184)
=2)&(Relative(169)>80)&(Relative(184)>30))|((ordinal(184)=1)&(ordinal(169)=2)&(Relative(184)
>80)&(Relative(169)>50));

**Section S3: Uncertainty of quantification of compounds with no authentic standards**

To assess the uncertainty of this method, we estimated known concentrations of compounds for which authentic standards were available. Table S3-1 shows the difference between concentrations estimated with the generic standard and the authentic standard. The overall uncertainty is difficult to estimate. This depends upon both the uncertainty associated with quantifying an individual compound (U), and the number of compounds in a polygon (n). Then

$$U \text{ (polygon)} = (U_1^2 + U_2^2 + U_3^2 \dots\dots U_n^2)^{1/2}$$

where $U_1$, $U_2$ …… $U_n$ are the uncertainties associated with individual compounds. There were three polygons for which calibration standards were available for all compounds. These gave collective uncertainties calculated as above of 13.9%, 18.9% and 39.3% (mean = 24%). Although statistically rigorous, we feel that this overestimates the uncertainty as the mass closure figures for the samples as a whole appear realistic and none deviates appreciably from 75-100%, including samples not described in this paper.

**Table S3-1.** Comparison of true calibrated concentrations and estimated concentrations using this
methodology.

| Compound | True Calibrated Concentrations in sample (ng/µL) | Concentration using TIC and n-alkane calibration (ng/µL) | % difference |
|---|---|---|---|
| Pristane | 9.9 | 9.3 | -6% |
| Phytane | 10.0 | 10.4 | 4% |
| Cyclohexane, pentyl- | 3.1 | 3.3 | 6% |
| Cyclohexane, hexyl- | 15.2 | 18.1 | 19% |
| Cyclohexane, heptyl- | 14.0 | 16.5 | 18% |
| Cyclohexane, octyl- | 11.1 | 12.5 | 13% |
| Cyclohexane, nonyl- | 8.9 | 10.0 | 12% |
| Cyclohexane, decyl- | 10.0 | 9.5 | -5% |
| Cyclohexane, undecyl- | 3.8 | 4.0 | 5% |
| Cyclohexane, dodecyl- | 4.1 | 3.8 | -7% |
| Cyclohexane, tridecyl- | 3.7 | 3.7 | 0% |
| Cyclohexane, tetradecyl- | 3.1 | 2.5 | -19% |
| Cyclohexane, pentadecyl- | 3.0 | 2.8 | -7% |
| Compound | True Calibrated Concentrations in sample (ng/µL) | Concentration using TIC and n-alkylcyclohexane calibration (ng/µL) | % difference |
| naphthalene, 1-methyl- | 20.0 | 24.0 | 20% |
| naphthalene, 1-ethyl- | 15.0 | 13.0 | -13% |
| naphthalene, 1-propyl- | 12.0 | 10.0 | -17% |
| naphthalene , 1-hexyl- | 32.0 | 35.0 | 9% |

[Figure]

**Figure S4-1.**

[Figure]

**Figure S4-2.** TIC intensity response to the molar quantity (left) and mass(right) of individual n-alkanes and n-alkyl cyclohexanes (C$_{25}$-C$_{34}$).

[Figure]

**Figure S4-3.** TIC intensity response to the normalized molar quantity of individual deuterated standards (C$_{12}$, C$_{15}$, C$_{20}$, C$_{25}$, C$_{30}$).

[Figure]

**Figure S4-4.**

**Section S5S4: Representative Mass Spectra for diesel and gas phase hydrocarbon**

**identification**

Experimentally determined Representative representative mass spectra at 70eV and 12eV ionisation are shown below. These mass spectra were identical at their respective retention times in different samples. Mass spectra at the bottom panel (blue peaks) are from 12 eV ionisation, while the top panels (red peaks) are from 70eV ionisation.

[Figure]

**Figure S5S4-1.** Representative mass spectra of n-alkanes, n-alkylcyclohexanes and n-alkylbenzenes.
Top graphs with red peaks are 70eV ionisation mass spectra, while bottom graphs with blue peaks
are 12eV spectra.

[Figure]

**Figure S4-2.** Representative mass spectra of n- alkylnaphthalenes, cis and trans-decalin, tetralin, methyl-, dimethyl-, trimethyl-, and tetramethyl tetralins. Top graphs with red peaks are 70eV ionisation mass spectra, while bottom graphs with blue peaks are 12eV spectra.

[Figure]

**Figure S5S4-3.** Representative mass spectra of four PAH. Top graphs with red peaks are 70eV
ionisation mass spectra, while bottom graphs with blue peaks are 12eV spectra.

When two compounds co-elute, it is often possible to quantify each by using the low ionisation energy mass spectrum.   The molecular ion of the higher molecular weight component is used to provide an ion current which is scaled upward to include the other peaks in the mass spectrum to give a total ion current due to the component.   The second component can be determined by difference. The molecular ions for a $C_{11}$ and a $C_{12}$ cycloalkane are at m/z 154 and 168, respectively.

These peaks would be present in the mass spectrum of co-eluting compounds, but can be separated by isolating the ion current attributed to a peak by assessing the mass spectrum that contains only the molecular ion of one of the compounds. This avoids the issue of co-elution of some compounds.

[Figure]

[Figure]

[Figure]

[Figure]

[Figure]

[Figure]

For the diesel fuel 90% of the total mass injected was accounted for. We suspect that a significant amount of the mass that was unaccounted may be <$C_{10}$ and/or any unresolved peaks that we were unable to measure and/or identify using our technique. There were 123 unresolved peaks in the 2D chromatogram and their positioning is shown in Figure S6-1. The intensities of the peaks, however, are relatively small and thus suggest a small contribution to the overall mass. Representative mass spectra of some of these unresolved peaks are shown in Figure S6-2.

**Figure S6-1.** A 2D chromatogram showing the positioning of unresolved peaks (red oval).

**Figure S6-2**. Representative mass spectra of unresolved peaks by this methodology. These unresolved peaks were present in the chromatographic space outlined in Figure S6-1.

[Figure]

Figure S5. A chromatogram of diesel exhaust emissions in the gas phase collected on an adsorption tube and analysed via thermal desorption with 70eV ionization mass spectrometry.

**Section  S6 – Representative mass spectra for compound classes identified in the lubricating**

**oil**

n + i-Alkanes were identified by the SIC of m/z 57 and selecting the corresponding molecular ion of the alkane. Figure S8-1 shows two representative alkanes with the molecular ions 282 (C20) and (C21).

[Figure]

**Figure S8S6-1.** n-Alkane mass spectra at 12eV for m/z 57 and the molecular ion 282 and 296 for
C20 n-alkane (top mass spectum, red) and C21 n-alkane (bottom mass spectrum, blue), respectively.

Monocyclic alkanes were identified by the SIC of m/z 82 and selecting the corresponding molecular ion of the specific monocyclic alkane. For example, Figure S8-2 shows two representative monocyclic alkanes with the molecular ions 280 (C20) and 294 (C21).

[Figure]

**Figure S8S6-2.** Monocyclic alkane mass spectra at 12eV for m/z 82 and the molecular ions 280 and
294 for C20 monocyclic alkane (top mass spectum, red) and C21 monocyclic alkane (bottom mass
spectrum, blue), respectively.

Bicyclic alkanes were identified by the SIC of m/z 137 and selecting the corresponding molecular ion of the specific bicyclic alkane. For example, Figure S8-3 shows two representative bicyclic alkanes with the molecular ions 292 (C21) and 306 (C22).

[Figure]

**Figure S8S6-3.** Bicyclic alkane mass spectra at 12eV for m/z 137 and the molecular ions 292 and
306 for C21 bicyclic alkane (top mass spectum, red) and C22 bicyclic alkane (bottom mass
spectrum, blue), respectively.

Tricyclic terpanes were identified by the SIC of m/z 191 and selecting the corresponding molecular ion of the specific tricyclic terpane carbon number. For example, Figure S8-4 shows two representative tricyclic terpanes with the molecular ions 276 (C20) and 290 (C22).

[Figure]

**Figure S8S6-4.** Tricyclic terpanes mass spectra at 12eV for m/z 191 and the molecular ions 276 and
290 for C20 tricyclic terpane (top mass spectum, red) and C21 tricyclic terpane (bottom mass
spectrum, blue), respectively.

This analysis was repeated for all the compounds shown in Table 1 and representative mass spectra are shown in Figures S8-5 to S8-7 below.

[Figure]

**Figure S8S6-5.** Monocyclic aromatics mass spectra at 12eV for m/z 92 and the molecular ions 288
and 302 for C21 (top mass spectum, red) and C22 (bottom mass spectrum, blue), respectively.

[Figure]

**Figure S8S6-6.** Pentacyclic alkanes mass spectra at 12eV for the molecular ions 258 (top mass
spectum, red) and methyl pentacyclic alkane with molecular ion 272 and m/z 257 mass fragment
(bottom mass spectrum, blue).

[Figure]

**Figure S8S6-7.** Hexacyclic alkanes mass spectra at 12eV for the molecular ions 298 (top mass
spectum, red) and methyl hexacyclic alkane with molecular ion 312 and m/z 297 mass fragment
(bottom mass spectrum, blue).

**Figure S7**

[Figure]

**Figure S7.** A selected ion chromatogram of lubricating oil (5W30) with mass fragments **(A)** m/z 92 and 119 signifying monocyclic and methyl-monocyclic aromatics (at 70eV ionization mass spectrometry) and **(B)** m/z 302 signifying the $M^+$ for $C_{22}$ monocyclic aromatic isomers (at 12eV ionization mass spectrometry).

**References**

Reichenbach, S. E.; Kottapalli, V.; Ni, M.; Visvanathan, A.: Computer language for identifying
chemicals with comprehensive two-dimensional gas chromatography and mass spectrometry.
J Chromatog A, **2005,** 1071, , 263-269, 2005.

---

## Author Comment (AC2) · 2 Mar 2018

The comment was uploaded in the form of a supplement:
https://www.atmos-meas-tech-discuss.net/amt-2017-366/amt-2017-366-AC2-supplement.pdf